# Technical note: A fast and reproducible autosampler for direct vapor equilibration isotope measurements

Jonas Pyschik[1], Stefan Seeger[2,1], Barbara Herbstritt[1], and Markus Weiler[1]

[1]Hydrology, Faculty of Environment and Natural Resources, University of Freiburg, Freiburg im Breisgau, Germany
[2]Division of Soil Physics, Department of Crop Sciences, Georg-August University of Göttingen

**Correspondence:** Jonas Pyschik (jonas.pyschik@hydrology.uni-freiburg.de)

**Abstract.**

To investigate water movement in environmental systems, stable isotopes ($^2$H and $^{18}$O) ratios of water are commonly used tracers. Analyzing the isotopic ratios of water in or adsorbed to substances like soil or plant tissue necessitates extraction or equilibration of water prior to analysis. One such method, direct vapor equilibration, is popular due to its cost-effectiveness and straightforward sample processing. However, sample analysis requires significant manual labor, thereby limiting the number of samples that can be analyzed. This limitation is compounded by the fact that stored samples undergo evaporative isotopic changes over time, and in addition manual measurements require many laborious procedural steps that can easily compromise reproducibility. The operator has to decide subjectively if the measurements are stable, and then record the analyzer readings. To address these challenges, we have developed a system that automates the analysis process. Our vapor autosampler for vapor samples, named VapAuSa, features a modular design allowing up to 350 ports for direct vapor equilibration samples. These ports sequentially connect the prepared samples to a laser isotope analyzer, enabling continuous automated measurements. Within the accompanying software, measurement criteria can be specified, facilitating reproducible analysis. The developed system was tested by co-measuring 90 soil samples and 21 liquid water samples of known $\delta$-values. VapAuSa measurements have a negligible measurement bias ($\delta^2$H and $\delta^{18}$O both $< 1 \times 10^{-13}$ ‰) and similar measurement repeatability compared to manual analysis of identical samples (VapAuSa $\delta^2$H= $\pm$ 4.5 ‰ $\delta^{18}$O = $\pm$ 0.58 ‰ vs. manual $\delta^2$H= $\pm$ 5.7 ‰ $\delta^{18}$O = $\pm$ 0.37 ‰). However, the increased sample throughput minimizes storage-induced isotopic changes. Moreover, VapAuSa triples sample throughput per week while reducing direct labor time to just 10 % of manual processing.

## 1 Introduction

Stable water isotopes ($\delta^{18}$O and $\delta^2$H) have found widespread application as tracers in earth and environmental system sciences. They are applied to elucidate storage and redistribution processes of water in various hydrological and hydrogeological compartments. In soils, stable water isotope analysis revealed different flow-process along hillslopes like lateral- and preferential-flow as well as mixing (Garvelmann et al., 2012; Thomas et al., 2013; Peralta-Tapia et al., 2015). They were also used to estimate soil properties by inversely modeling soil-water isotope profiles, better representing the soil properties than traditional pedotransfer functions (Sprenger et al., 2016). In addition, the tracking of vertical infiltration using soil isotope profiles has

allowed for the quantification of groundwater recharge rates (Filippini et al., 2015; Chesnaux and Stumpp, 2018; Boumaiza et al., 2020). Analyzing the ratios of stable water isotopes in groundwater has revealed hydrogeological differences and solute transport mechanisms (Hendry and Wassenaar, 2009; Hendry et al., 2011a; Hendry and Wassenaar, 2011; Stumpp and Hendry, 2012). Moreover, the use of stable water isotopes in plants has shed light on plant water uptake, water transit times, and water partitioning, providing valuable insights into which sources of water plants utilize for various purposes (Bertrand et al., 2014;

Smith et al., 2020; Kuhlemann et al., 2020).

    To measure the stable water isotope ratios of water bound to substances or tissues, such as soil, sediment or plant material, an extraction or equilibration to vapor is needed. One such method is direct vapor equilibration laser spectroscopy (DVE-LS) (Wassenaar et al., 2008). In DVE-LS, the water vapor of a sample is measured and recalculated to its liquid isotope ratios using calibration standards. The process works as follows: (1) The sample (soil, plant, etc.) is placed in an inflatable, sealable and

gas diffusion-tight bag (Pratt et al., 2016). Many different bag-materials have been used, however most are insufficient due to diffusive losses. These losses can be minimized with aluminum laminated bags (Gralher et al., 2021).Regarding identical treatment of calibration standards (Werner and Brand, 2001), liquid water of known isotopic composition is also filled into identical bags. (2) The sample and calibration standard bags are inflated with dry air and sealed. Then, a silicone blot is added to each bag as a septum, needed for airtight measurements later. (3) Samples are then stored under constant climatic conditions

(ideally in a air-conditioned room where they are later analyzed) so liquid and gas phase within the bags can reach isotopic equilibrium (Wassenaar et al., 2008). Equilibration time varies between studies but for aluminum laminated bags, 48 hours are optimal for soil samples (Gralher et al., 2021). (4) After equilibration, the vapor in the bag's headspace is analyzed using off-axis integrated cavity output spectroscopy (OA-ICOS) or cavity ring-down spectroscopy (CRDS). To facilitate this analysis, a cannula connected to the analyzer's inlet port is inserted into the silicone septum of the bag. Maintaining thermal stability is

critical during both equilibration and analysis, as the fractionation of stable water isotopes is highly temperature dependent. (5) After analysis, the measurements are normalized to the VSMOW-SLAP scale using the co-measured calibration standards (Craig, 1961; Pratt et al., 2016).

    Advantages of DVE-LS compared to other extraction methods include low technical effort and minimal material requirements, making it a cost-effective option (Millar et al., 2018). Additionally, only a small sample volume is necessary, allowing

for high spatial and temporal resolution sampling (Wassenaar et al., 2008; Garvelmann et al., 2012). Furthermore, minimal handling is required, reducing the risk of sample damage (Wassenaar et al., 2008). Due to these favorable attributes, DVE-LS finds application in various contexts. For instance, all findings presented in the first paragraph were obtained through the application of DVE-LS. The wide application of DVE-LS led to numerous methodological improvements, most focused around container material, equilibration time (Gralher et al., 2021), the correction of variations of the carrier gases (Gralher et al.,

2016, 2018) or co-extracted substances which interfere with the laser spectrometer analysis (Hendry et al., 2011b).

    However, the method still lacks automation in the analysis process. Manual measurement of each sample is time-consuming, imposing limitations on high number sampling, as sample storage can alter the isotopic values due to evaporation and diffusion (Gralher et al., 2021). Moreover, the current DVE-LS analysis routine lacks reproducibility. There is no standardization for analysis time and measurement stability criteria, leading to potential variations in results among different labs and opera-

60 tors analyzing the same sample (Millar et al., 2022; Ceperley et al., 2024). To address these challenges and enhance both the speed and objectivity of the analysis, we introduce our fully automated vapor sampler system, named "VapAuSa". This innovative system enables high-throughput sample processing with significantly reduced manual labor compared to the prevailing procedure.

## 2  Design

This section provides a summary of the sampler design. More in-depth information such as the list of materials, technical drawings, circuit diagram, code, and manuals, necessary for building the VapAuSa, can be found in the Supplement and on https://gitlab.rz.uni-freiburg.de/hydrology/vapausa.

### 2.1  Hardware

The VapAuSa features a modular design comprising standalone boxes holding the sample bags that can be combined to larger
sample setups. Each box is equipped with 24 ports, of which 23 are designated for sample bags, and one is reserved for flushing the system with ambient air to prevent moisture build-up and thus memory effects. The ports consist of valves (Model E3O10A-1W024; Clippard) which regulate vapor flow to the cavity of the ring-down isotope analyzer (in our case a L2130-*i*, Picarro Inc.). The valves are attached to CNC-cut aluminum valve-blocks organized in groups of eight and interconnected by 1/8 inch PTFE tubing (Figure 1). The 23 sample bags are connected with cannulas (2.1 mm diameter * 80 mm, B.Braun STERICAN)
to the valves through 1/8 inch PTFE tubing. The tubes are secured by super glue (LOCTITE 406 and 770, Henkel Adhesives) into the cannula and linked to the valve block via flangeless fittings (XP-301X; IDEX) (see Figure 1). We experimented with using 1/16 inch tubes and a gas-drying cartridge at the flushing valve, but this did not yield improved measurements.

### 2.2  Circuit board, electronic components and Firmware

The first box of each VapAuSa is controlled by the primary module, while all subsequent boxes are controlled by extension
modules. We have designed a circuit board that can be used to build either of these two configurations, depending on the electronic components installed. The primary and extension modules share most parts but are equipped with different types of micro-controllers and connectors. This distinction arises because only the primary module can directly be connected to the power supply and to the Picarro. Each extension module is equipped with a DIP-switch that can be used to assign a unique address between 1 and 16 (the primary module is assigned address 0). In total, this allows for up to 17 boxes in
one VapAuSa, capable of accommodating 391 samples simultaneously. The primary module and all extension modules can be daisy-chained one after another, with communication among them facilitated through a RS485-bus system. Communication and power supply for all extensions are achieved using off-the-shelf Ethernet cables. A detailed part list for the required electronic components to equip the circuit boards can be found in the appendix of this paper. Circuit boards schematics (created with Fritzing) and the firmware for the micro-controllers of primary board and extensions can be found under https://gitlab.rz.uni-
freiburg.de/hydrology/vapausa.

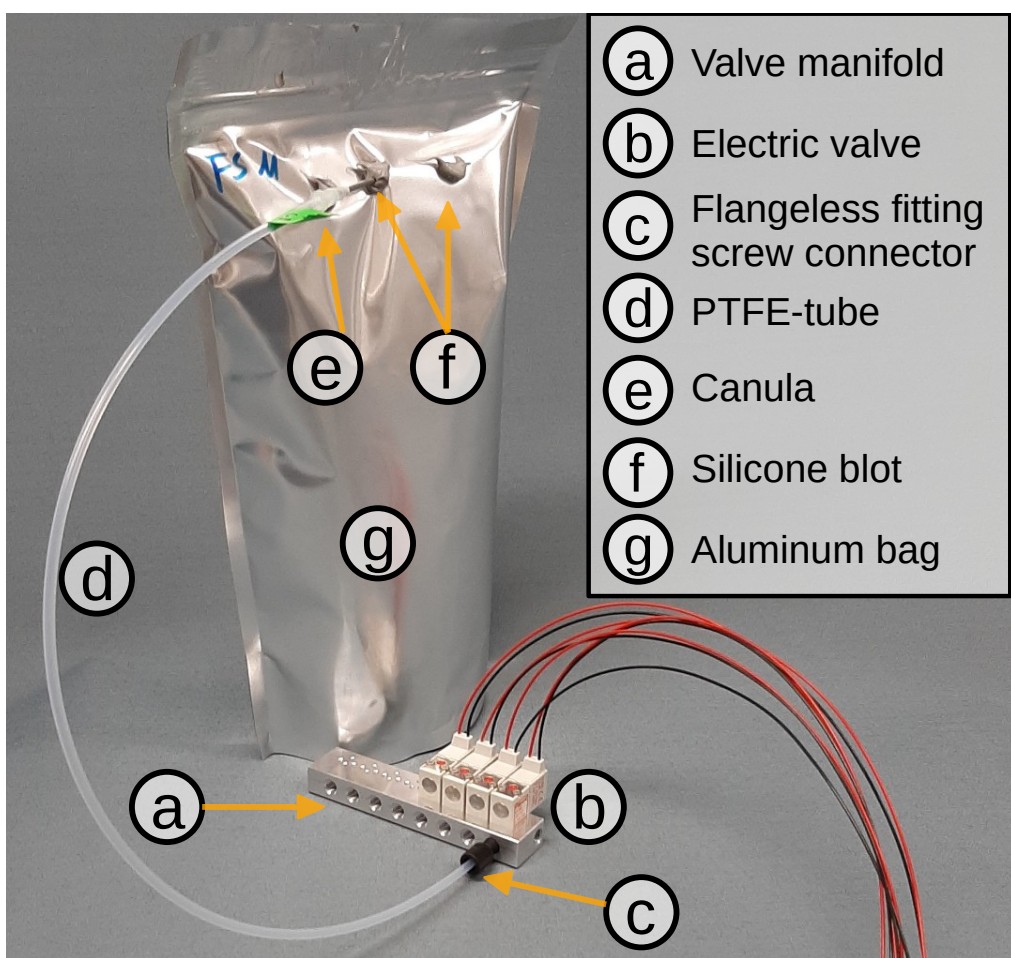

**Figure 1.** Basic setup of the VapAuSa: A cannula ⓔ is superglued onto a PTFE-tube ⓓ, which is connected to a CNC-cut valve manifold ⓐ via a flangeless fitting connector (XP-301X; IDEX) ⓒ. In this example image, only four of the eight positions on the valve manifold are equipped with electric valves (E3O10A-1W024; Clippard) ⓑ. The cannula is inserted to the laminated aluminum bag ⓖ via a silicone blot ⓕ, which acts as a seal after the bag was been pierced by the cannula.

## 2.3 Software

The software used for the VapAuSa is based on software that was developed for in-situ stable water isotope sampling (Seeger & Weiler, 2021). It was designed to operate directly on a Piccaro stable water isotope analyser (model L2120-*i*, L2130-*i* or L2140-*i*) and consists of a collection of modular Python3 (Python-Software-Foundation, 2022) scripts whose purpose is to
fulfil two main tasks:

*a) Monitoring the measurements*: By frequently reading out and parsing the instrument's log-files, the software is aware of the currently measured values of sample gas volumetric vapor content ($H_2O$), $\delta^{18}O$ and $\delta^2H$ (see ④ in Figure 4). Through a

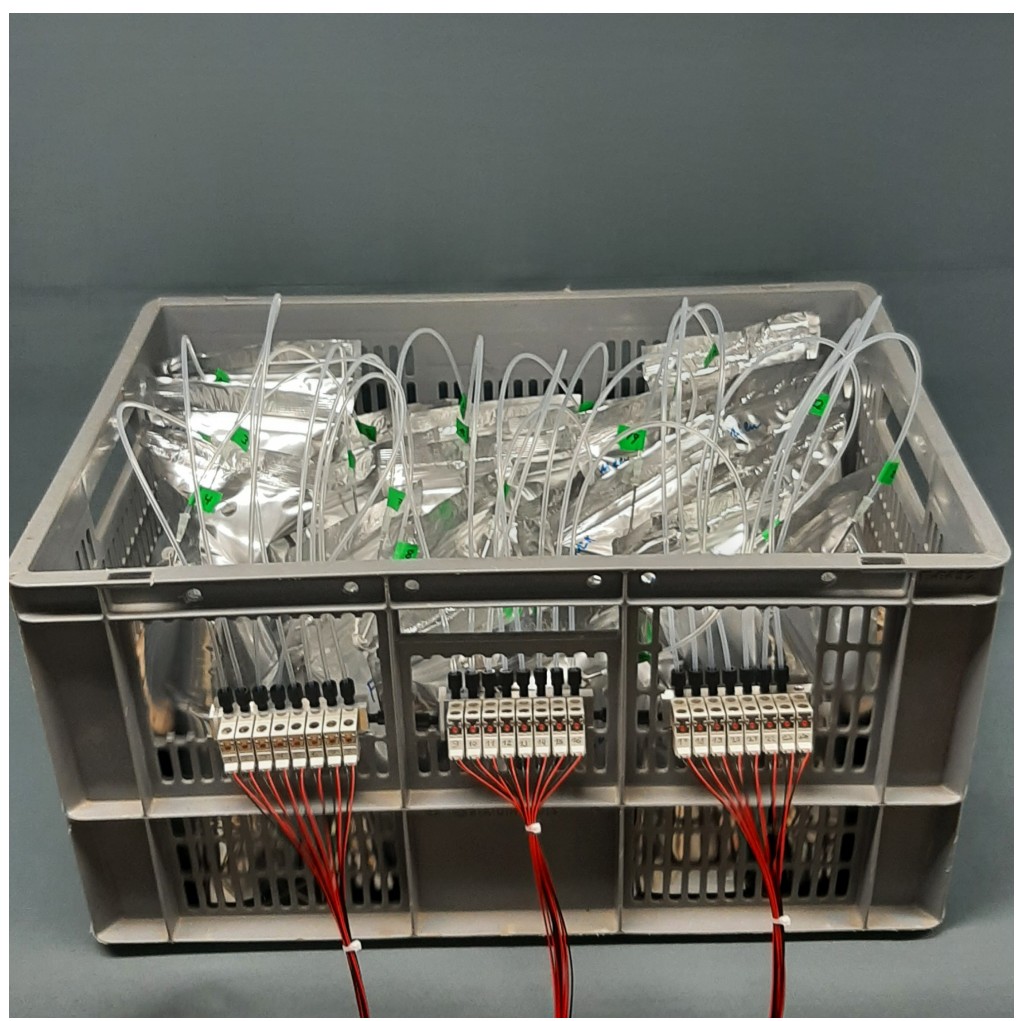

**Figure 2.** VapAuSa standalone box holding the 23 sample bags with the 24 valves

configuration file, the user can define stability criteria, specifying trends and standard deviations of the mentioned measurement parameters over a user-defined time span (⑤ in Figure 4). This establishes an objective metric for the automated detection of
100 stable plateaus during measurements. This module can operate independently and is potentially suitable for manual DVE-LS measurements, since it provides instant one-click summary statistics for user-selectable time spans. In contrast, the standard Picarro analyzer GUI often requires tedious zooming in and out to achieve the same purpose.

*b) Controlling the valves and sampling*: Using another configuration file, the user can assign custom names (sample IDs) to specific valve slots. Each valve slot is uniquely addressed by a combination of boxID and slotID (e.g., 2#5 addresses the
105 fifth valve slot of the second extension box). The user has the flexibility to set a maximum sampling time for each valve and determine whether sampling may conclude before that time if all stability criteria are met. Additionally, the user can establish

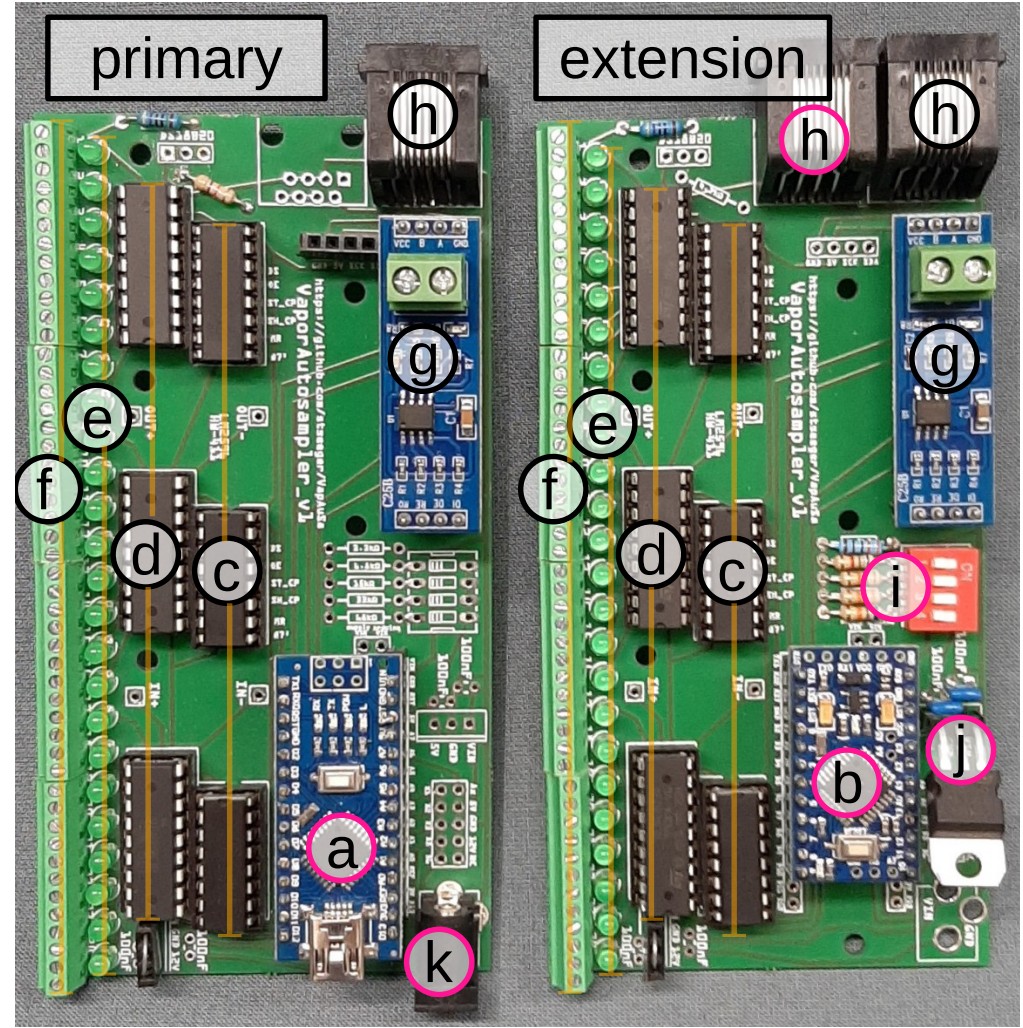

**Figure 3.** VapAuSa primary module (left) and an extension module (right). Based on identical circuit boards, differences in equipped components (pink circles) lead to the two kinds of modules used in the VapAuSa sytsem. ⓐ Arduino Nano, ⓑ Arduino ProMini, ⓒ Shift-Register, ⓓ ULN2803 TransisorArray, ⓔ Status LEDs, ⓕ Screw terminals, ⓖ RS485-Max communication module, ⓗ Ethernet port, ⓘ Address selection DIP-Switch, ⓙ 5V voltage regulator, ⓚ 12VDC power jack.

a custom sequence for all defined valve slots, enabling repeated measurements from specific slots (e.g., standards) within a single sequence. During an active sequence, each measurement phase is preceded by a flushing phase during which the valve block of the currently active slot is flushed with ambient air until a defined $H_2O$ concentration (ppm in the analysis chamber) is

110 undershot or a maximum flush time has been reached. Upon running the main script, a graphical user interface (GUI) visualizes the recent measurements (④ in Figure 4). This GUI also allows the user to manually open certain valves by clicking on the corresponding buttons (① in Figure 4). The GUI automatically starts the predefined sequence (③ in Figure 4) and generates

a log file that documents each valve switch and the reason why it occurred (timeout, fulfillment of all stability criteria, or manual click). By combining the analyzer log file with the valve log file, it is possible to automatically aggregate and process

the relevant parts of the analyzer's measurements. The software identifies a connected VapAuSa primary module by scanning all available Serial-COM ports and then sends the appropriate `valve boxID#slotID` commands to the primary module (firmware described in Sec. 2.2) in response to user inputs or progression through the predefined sequence. An example call of the `valve` command might be `valve 2#5`, which means that the fifth valve of the second extension box should be opened. The receiving primary module would relay this command to all connected extensions. Subsequently, all boxes that are not the

second extension would close all of their valves and the second extension would open its fifth valve.

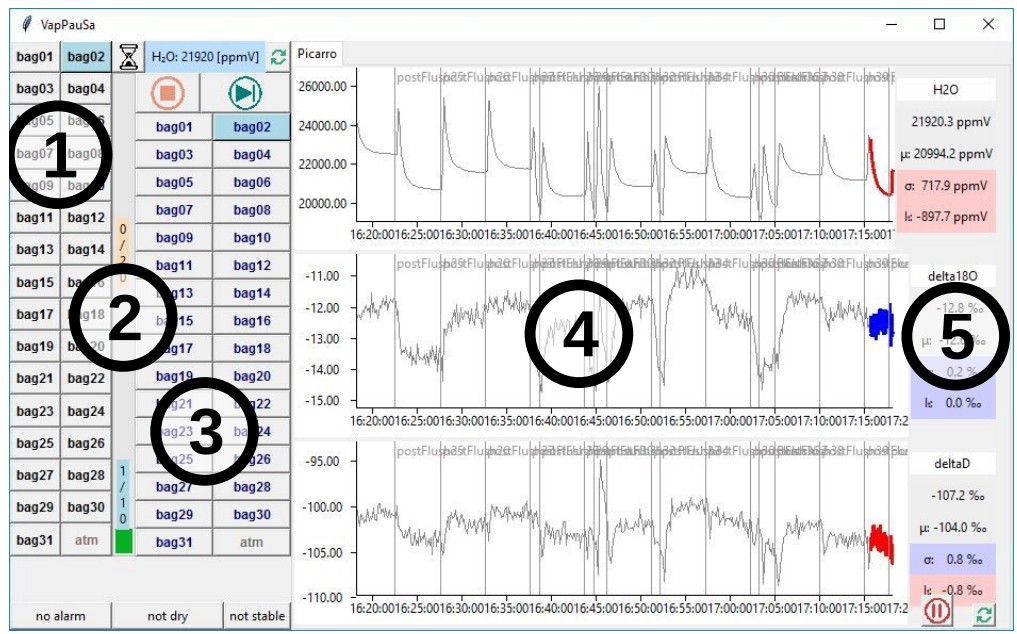

**Figure 4.** Screenshot of the VapAuSa GUI. ① Buttons for all valves defined in the configuration file, ② Flush and measurement progress bars for currently active valve, ③ Buttons for valve sequence, ④ Recent measurement values for $H_2O$, $\delta^{18}O$ and $\delta^2H$, ⑤ Fulfillment of stability criteria (red=not fulfilled, blue=fulfilled) for $H_2O$, $\delta^{18}O$ and $\delta^2H$

## 3    Proof of Concept

### 3.1    Soil Water Test

To evaluate the effectiveness of the VapAuSa for soil samples, we performed a comparative study involving both manual and autosampler measurements of identical soil sampling bags. We drilled 11 soil cores to refusal depth using an electrical auger

(Makita HM1810), resulting in a total of 90 soil samples of different depths. These samples (2-4 tablespoons) were then placed into aluminum-laminated plastic bags (WEBER Packing GmbH; CB400-420BRZ; 500 ml) and initially sealed with a ziplock.

**Table 1.** Stability criteria for the VapAuSa measurements. Only if standard deviation (sd) and trend are all below the given thresholds within evaluation time (eval. time), VapAuSa switches to the next sample.

|  | sd [units] | trend [unit/eval. time] | eval. time [s] |
|---|---|---|---|
| $H_2O$ [ppmv] | 100.0 | 150.00 | 120 |
| $\delta^{18}O$ [‰] | 0.2 | 0.08 | 120 |
| $\delta^2H$ [‰] | 0.7 | 0.30 | 120 |

Afterwards, the bags were inflated with dry air, heat-sealed, and equipped with two silicone blots to ensure each measurement started with a "fresh" septum. The prepared bags were then stored in the climate-controlled analysis room maintained at 20°C ± 1°C for 48 hours. All measurements were performed on cavity ring-down spectrometers (L2120-*i* and L2130-*i*, Picarro Inc.). The bags were punctured with a cannula and connected to the analyzers. Standards of known isotopic ratios were co-measured to calibrate the results. Additionally, the concentration of $H_2O$ in the analysis chamber was corrected to account for fractionation caused by temperature changes. To achieve this, a linear regression was performed between the $H_2O$ concentration (in ppm) of the standards and the corresponding differences in isotope values. Using this regression, each measurement was adjusted to a standardized $H_2O$ concentration of 25000 ppm, ensuring consistency across the data. To compare the measurement methods, 12 bags were measured manually first and then by the autosampler, while the remaining were measured by the autosampler first and then manually. The VapAuSa was programmed to activate valves for 10 minutes each, with 5 minutes of system flushing between samples. Stability criteria were defined as shown in table 1, which represent the threshold after which the VapAuSa switches to the next sample. To assess the measurements, we calculated the difference for each bag by subtracting the calibrated $\delta$ values measured by VapAuSa from the calibrated $delta$ values measured manually.

## 3.2 Liquid Water Test

Since the real isotopic ratios of soil water is currently impossible to determine (Koeniger et al., 2011; Orlowski et al., 2013, 2016; Gaj et al., 2016), we assessed the measurement repeatability and bias of the VapAuSa by measuring different liquid DVE-LS samples. The test samples were created according to the suggested protocol by Wassenaar et al. (2008) and Gralher et al. (2021). Seven samples (10 ml each) of three isotopically distinct water sources were filled into 1 L aluminum laminated plastic bags. Subsequently, we inflated the bags with dry air, heat-sealed and equipped them with silicone septa. Then the bags were placed in a climate controlled room (20°C ± 1°C) for 48 hours. Following this period, we randomly distributed the samples in the VapAuSa to minimize possible memory effects. To connect the bags to the isotope analyzer (cavity ring-down; L2130-*i*, Picarro Inc.), we inserted the cannulas into the septa, puncturing the bags and enabling vapor flow to the analyzer. The VapAuSa was programmed identical to the soil water tests with the stability criteria shown in table 1. To test a wide isotopic range we selected water samples with three distinct compositions: relatively high $\delta$ values (sea-water, $\delta^{18}O$ = 0.71 ± 0.04 ‰ $\delta^2H$ = 0.27 ± 0.26 ‰), medium $\delta$ values(tap-water of Freiburg, $\delta^{18}O$ = -9.31 ± 0.04 ‰ $\delta^2H$ = -64.33 ± 0.26 ‰) and low $\delta$ values (snow-melt water, $\delta^{18}O$ = -16.62 ± 0.04 ‰ $\delta^2H$ = -125.84 ± 0.26 ‰). These reference isotope ratios

were derived by measuring sub-samples on a liquid isotope analyser (L2130-*i* with attached vaporizer unit A0211, Picarro Inc.) and are considered the samples "true" value throughout the analysis. After the samples were analysed on the VapAuSa, we

calibrated the results to these liquid measurements.

To quantify the deviations of the VapAuSa measurements, the results ($\delta$-values VapAuSa) were subtracted by the respective liquid analyzer results ($\delta$-values liquid). Measurement bias was then calculated as the mean deviation of all samples. We determined the measurement repeatability as one standard deviation (1 $\sigma$) for each of the three water sample groups. We also assessed the VapAuSa performance by comparing the measurement variability of 230 manual measurements from identical wa-

ter sources. There we again calculated the deviations as the manual measurements ($\delta$-values hand) subtracted by the respective liquid analyzer results ($\delta$-values liquid).

### 3.3    Results

#### 3.3.1    Soil Samples

The distribution of the soil water measurements showed good overall agreement. Both manual and automatic measurements

had similar means and standard deviations, indicating consistent results. Additionally, both boxplots intercept 0, ruling out systematic measurement errors. This suggests no bias attributable to the VapAuSa (Figure 5). Although the standard deviation was relatively high, it is similar across the measurements and suggests that the variance is likely due to the bags being measured twice, a theme that will later be discussed. Specifically, bags sampled first by the autosampler and then manually had a lower standard deviation (0.96 in $\delta^{18}$O and 6.25 in $\delta^2$H) compared to those measured manually first (1.44 in $\delta^{18}$O and 9.14 in

$\delta^2$H). Additionally, the coefficient of determination varied depending on the initial measuring device: when the soil water was measured by hand first and then by the autosampler, the $R^2$ for the evaluated linear relationship between the autosampler and hand measurements was a low 0.31. However, if the autosampler measured first, the $R^2$ increased to 0.71 (Figure 6).

There is a clear divergence of the $R^2$ from soil bags measured first manually versus first with the autosampler. The soil samples which were measured first by the autosampler were often measured directly afterwards manually, while the first

manually measured samples were placed in the autosampler and measured up to six hours later due to their positioning in the system. There is always a risk that once a bag is punctured and the cannula is pulled out, the seal might not be tight, potentially leading to higher uncertainty with the hand-first bags due to the extended time allowed for leakage through the pinched septum. Since the uncertainties among bags measured first manually and those measured by the autosampler are similar, this suggests that the largest uncertainty is induced by measuring the bags twice. If the VapAuSa would decrease measurement precision,

this would manifest as a consistent bias, either always increasing or decreasing the $\delta$ values.

#### 3.3.2    Water Samples

The measurement bias (mean difference of VapAuSa measurement to liquid value) was extremely low; -3.9 $\times 10^{-15}$ ‰ for $\delta^{18}$O and 7.1 $\times 10^{-14}$ ‰ for $\delta^2$H. Average repeatability across all samples was $\pm$ 0.58 ‰ for $\delta^{18}$O and $\pm$ 4 ‰ for $\delta^2$H. However across the different water-sources, VapAuSa showed differing repeatabilities. The largest repeatability range was

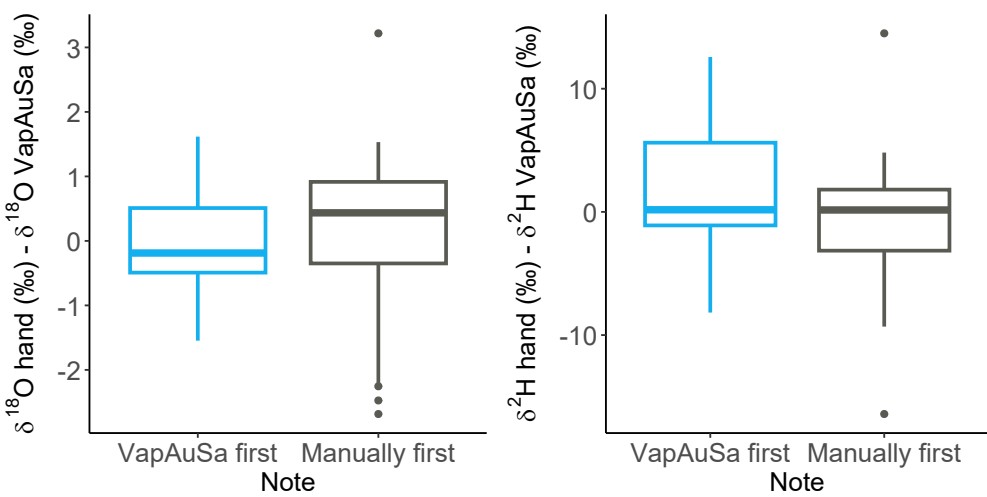

**Figure 5.** Differences of samples measured manually and with VapAuSa, depending which method measured first.

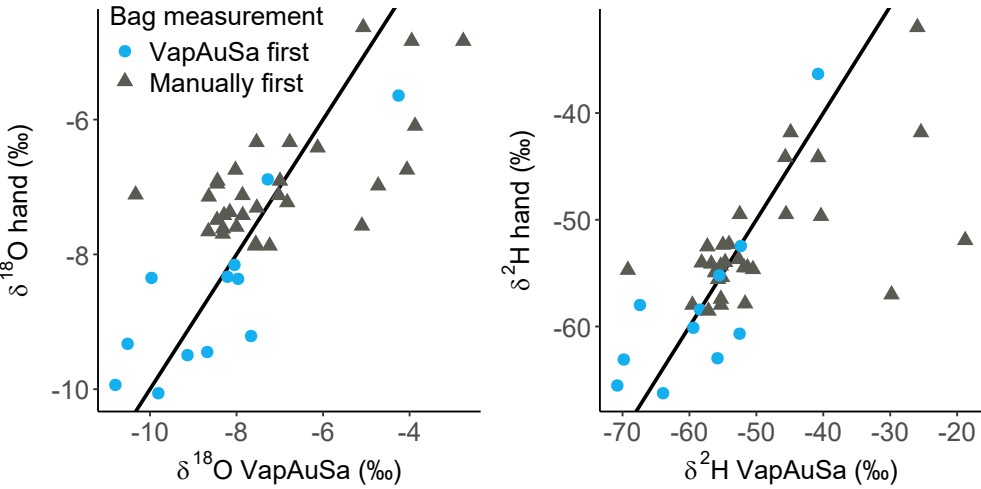

**Figure 6.** Relation of VapAuSa soil water measurements to manual measurements. 12 samples were measured by autosampler first, the rest manually first. The line represents a 1:1 relationship.

measured for the low $\delta$ value water with $\pm$ 0.82 ‰ for $\delta^{18}$O and $\pm$ 7.0 ‰ for $\delta^2$H. It performed best for the isotopic range of the medium and high $\delta$ values with similar deviations of around $\pm$ 0.47 ‰ for $\delta^{18}$O and $\pm$ 3.2 ‰ for $\delta^2$H (Figure 7).

Also 60% of the measurements stabilized before the evaluation time threshold. While we programmed the VapAuSa to analyze each sample for 10 minutes, most measurements met the stability criteria (Table 1) within 8 minutes. This reduction in overall runtime from 5 to 4 hours was observed across 21 sample measurements. The manual analysis showed a measurement repeatability (1 $\sigma$) of $\pm$ 0.37 ‰ for $\delta^{18}$O and $\pm$ 5.7 ‰ for $\delta^2$H, being in a similar range as those of the VapAuSa (Results

are shown in Figure 7). The increased uncertainty of the VapAuSa in the isotopically low range is also visible in manual measurements. While the manual repeatability range is 50 % lower in $\delta^{18}$O than that of the VapAuSa, it is 130 % higher for $\delta^2$H.

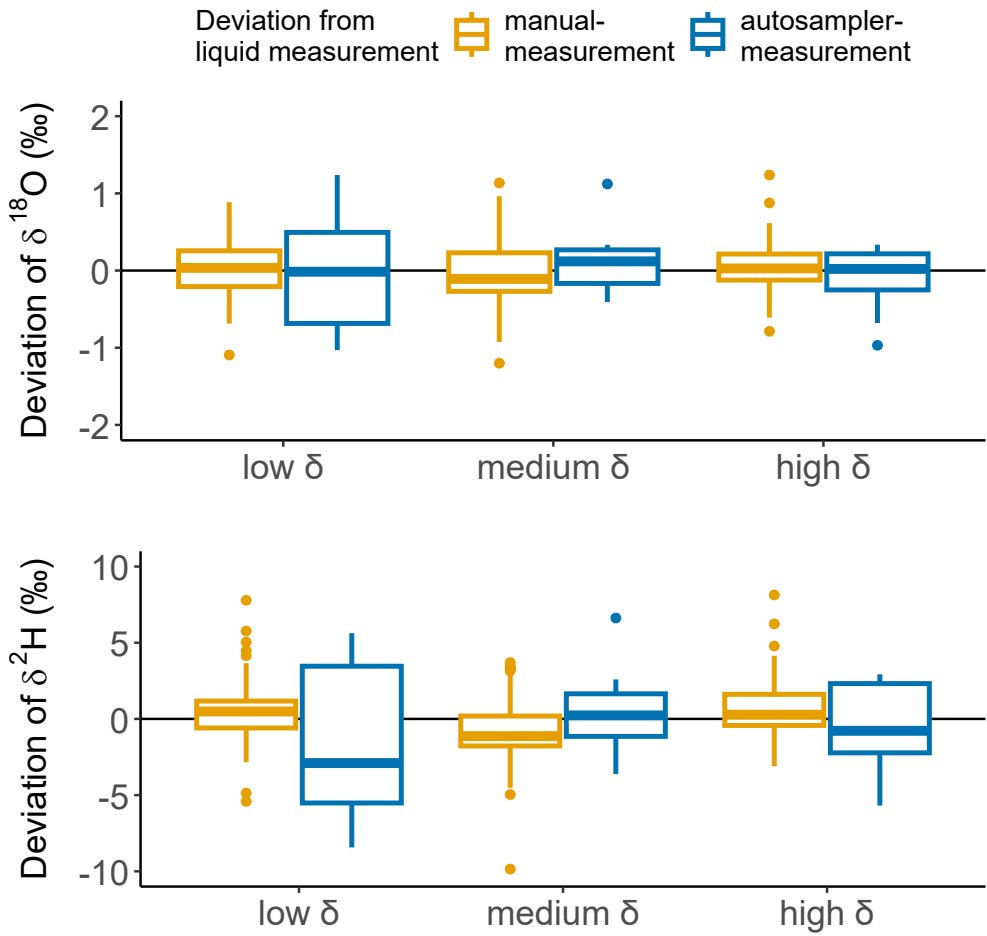

**Figure 7.** Deviation of VapAuSa and manual measurements from liquid measurements of 3 water samples of different isotope ratios

As observed with the two times measured soil samples, pinched bags can alter their isotopic ratios. Therefore, it is crucial to
evaluate the effect of the duration a sample remains in the autosampler, from the time it is placed and pinched by the cannula to the time of measurement. In the liquid water test, the time between pinching and measurement extended up to 4 hours. When analyzing the drift over time, the slopes for each bag measurement varied. The slopes of the water source with lower and higher $\delta$ values indicated an enrichment in heavier isotopes at rates of 0.31 ‰ per hour and 0.20 ‰ per hour, respectively, while the medium $\delta$ value water source showed depletion at a rate of -0.04 ‰ per hour. The slope for the high $\delta$ value is the only one

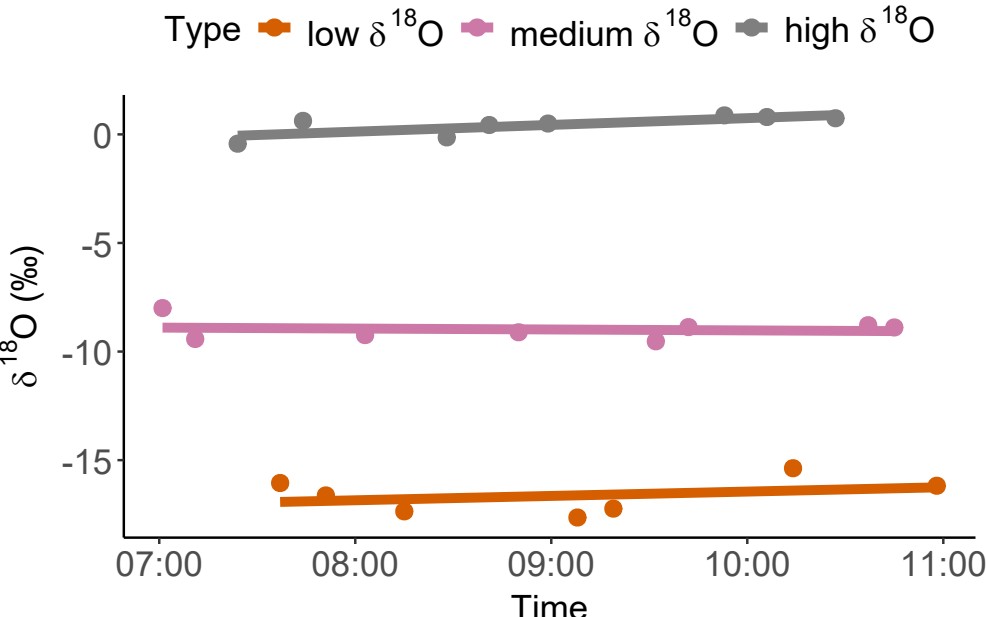

**Figure 8.** VapAuSa measurements of three water sources, each dot representing an individual bag. Isotopic drifts are indicated by the regression lines with slopes of 0.31 ‰ per hour for low $\delta$ values, -0.04 ‰ per hour for medium $\delta$ values and 0.20 ‰ per hour for high $\delta$ values.

that is statistically significant at $p<0.05$, while the slope of the medium and low $\delta$ water sources are not statistically significant. Therefore, no overall trend could be identified, suggesting that the drift is of minor importance to the measurement accuracy. However, this effect can always be analyzed and corrected for with the co-measured calibration standards.

## 4  Discussion

### 4.1  Accuracy

Oven dried soils spiked with water of known isotopic values are often used to examine extraction and equilibration performance (West et al., 2006; Wassenaar et al., 2008; Volkmann and Weiler, 2014). Accuracies reported for DVE-LS vary between 0.7-1.0 ‰ for $\delta^2$H and 0.2-0.3 ‰ for $\delta^{18}$O (Wassenaar et al., 2008; Volkmann and Weiler, 2014). However, the soil texture can alter the extracted isotopic compositions. While the $\delta$ values of extracted water from sandy soils is often close to the known isotope ratios, a high clay content changes the isotopic composition (Koeniger et al., 2011; Orlowski et al., 2013, 2016; Gaj et al., 210    2016). This clay-induced error depletes isotopic values in DVE-LS (Orlowski et al., 2016). Since a true accuracy assessment using spiked soil samples is currently impossible, we chose to evaluate the usability of VapAuSa for soil samples by comparing hand and autosampler measurements of natural soil samples.

In order to exclude soils as a source of uncertainty in testing the VapAuSa system measurement accuracies (both measurement repeatability and bias), we chose a water-to-water equilibration of known liquid samples. This approach has not been previously applied to DVE-LS, making its classification challenging. However, water-to-water extraction has been assessed in the context of cryogenic vacuum distillation, which can serve as a benchmark. In such assessments, water-to-water extraction exhibited a measurement repeatability of 0.2 ‰ for $\delta^{18}$O and 0.8 ‰ for $\delta^2$H (Orlowski et al., 2016). In comparison, the DVE-LS autosampler as well as the manual measurements demonstrated lower accuracy. But the VapAuSa offers advantages in higher sample throughput and reduced manual labor compared to the cryogenic method.

Because of the larger repeatability range, the applicability of the VapAuSa depends on the $\delta$-value ranges of the water source ratio differences of interest. This issue has been previously discussed in the context of the uncertainty introduced by cryogenic extraction (Allen and Kirchner, 2022). The conclusion was that two sources can be differentiated in a mixing model only if the $\delta$-value span between them is substantially larger than the uncertainty added by cryogenic extraction. This principle also applies to the VapAuSa. When the $\delta$-value range of the end members is small,another extraction method with lower uncertainties should be chosen. However, if the $\delta$-value range is substantially larger than the repeatability uncertainty, VapAuSa is a good choice as it increases sample throughput.

## 4.2  Storage Time

Sample storage leads to evaporation and diffusion induced isotopic drifts, which occur over time with every DVE-LS container material. Even aluminum laminated bags sealed by their ziplock (as they would be during storage) have an isotopic drift of 0.042 ‰ per day for $\delta^{18}$O at room-temperature (Gralher et al., 2021). To minimize these shifts one should analyse the samples as quick as possible and keep the samples cooled. Manual DVE-LS sample analysis is limited to about 180 samples per week. However VapAuSa can analyze up to 120 samples per day thus enabling up to 500 sample measurements per week. Relating the throughput to storage drifts, analyzing 400 DVE-LS samples (2 weeks of manual analysis vs. 3.5 days VapAuSa analysis) adds about 0.6 ‰ storage induced uncertainty to manual $\delta^{18}$O analysis , however only 0.17 ‰ storage induced uncertainty to VapAuSa $\delta^{18}$O measurements. Therefore, with larger sample volumes, the measurement uncertainty of VapAuSa may be lower than manual analysis due to shorter storage times.

## 4.3  Reproducibility

By defining (and documenting!) stability criteria, measurements are less dependent on person specific skills and preferences regarding the detection of a "stable" measurement. The subjective influence of the operator diminishes even further, as the system treats all samples equally, without fatigue after several hours of measurements. Also, the monitoring part of the VapPauSa-GUI can be used separately and does not require any other hardware than the isotopic analyzer itself. We think it might even help to improve manual DVE-LS measurements, as it provides instant one click summary statistics for freely defined time spans. Meanwhile the standard Picarro-analyzer GUI requires tedious zooming in and out for that purpose. On top of that, the possibility to define objective stability criteria should also help to reduce the a subjective component of manual DVE-LS measurements.

## 4.4 Don't buy cheap

To gain the mentioned VapAuSa benefits, we want to stress the importance of using appropriate, vapor-resistant and vapor-impermeable materials. Even a tiny amount of ambient air continuously leaking into the system can cause major errors with water vapor analysis. Our first prototype was build using valves designed for liquids with the connection between valve and cannula only plugged into each other and secured with Parafilm (PARAFILM® M). While this system costs a fraction of our current setup, it eventually developed a leak. This lead to ambient air mixing with the sample vapor, causing wrong measurements. Therefore, it is important to use adequate materials like gas-rated valves and secure all connections either by screw-fittings or superglue. Only this can ensure reliable measurements.

## 5 Conclusions

VapAuSa is a helpful system for all disciplines applying DVE-LS. While the VapAuSa comes with certain measurement uncertainties, those measurements are less prone to manual measurement errors, less labor intensive and have an increased throughput. In cases where the source $\delta$-value range is larger than VapAuSa uncertainty, the larger sample throughput can be a great benefit. So far we tested the system with liquid and soil samples only, however investigating plant samples (which have an even shorter maximum storage time) would be an interesting next step. Generally we think the VapAuSa is a valuable addition to the tool-sets of isotope geoscientists, enabling high number sampling and measurement, needed for advancing our understanding of environmental systems.

*Code availability.* All of the scripts that compose the VapAuSa-GUI can be found under https://gitlab.rz.uni-freiburg.de/hydrology/vapausa.

*Author contributions.* JP, SS and MW designed the experiment. SS developed the circuit boards and firmware of the auto sampler as well as the VapAuSa software. JP conducted the experiments and data analysis and wrote the first draft. BH analyzed the standard samples. SS, BH and MW contributed to writing the final manuscript.

*Competing interests.* At least one of the (co-)authors is a member of the editorial board of Hydrology and Earth System Sciences.

*Acknowledgements.* This research has been supported by the Deutsche Forschungsgemeinschaft (project no. 453746323) through the research unit FOR 5288: "Fast and Invisible: Conquering Subsurface Stormflow through an Interdisciplinary Multi-Site Approach".

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
