# Peer review of "Technical note: A fast and reproducible autosampler for direct vapor equilibration isotope measurements"

_EGUsphere, 2024_

## Author Response (AR1)

**General changes made**

- We took 90 additional soil samples and evaluated the system against them

- We changed the accuracy metrics to repeatability and measurement bias

- We excluded CVD from the manuscript

- We updated the wording according to Zachary Sharp's 2017 text

- We changed the structure of the Discussion and added many paragraphs, including one about the pinching duration

**Authors' responses to the comments of reviewer#1**

We appreciate your review and comments on our manuscript, "Technical note: A fast and objective autosampler for direct vapor equilibration isotope measurements". Your feedback is valuable to us, and we will make the recommended revisions accordingly. We provide detailed responses to each of your comments below.

Section 1: General Comments:

The developed automated sampling apparatus is a valuable improvement to the DVE methodology as it increases sample throughput in turn potentially limiting storage impacts on isotope ratio measurements. The possibility of decreased in-lab labour will also be of benefit to labs with large sample throughput needs. This apparatus is a valuable addition to the field and it's open source design principles using (mostly) market available components is to be commended. The inclusion of the script GUI_Picarro.py (in the supplemental repository "Software/Scripts/") for users of Piccaro CRDS which, is also of great use to users of that analysis approach given the increase in objectivity of measurement it can provide as well as providing the impetus to openly report on measurement parameters, even without the VapAuSa hardware.

Overall, I suggest mostly minor revisions regarding the use of appropriate terminology, the addition of citations for some strong statements contained in the manuscript which currently lack needed support, and some improvements to clarity. There are a few more pressing questions regarding testing for memory effects which need to be addressed. This concise manuscript provides a valuable new technology for users of DVE-CRDS and is appropriate for publication in HESS given the improvements I suggest in the specific comments below.

My review of the supplemental material containing the list of materials, technical drawings, circuit diagram, code, and manuals, necessary for building the VapAuSa" was cursory given my familiarity with some of the file types but the repository appeared to contain all of the required materials and information needed to construct the apparatus. I must admit the construction of such an apparatus falls outside of my expertise so I cannot be fully certain of any missing components, but the repository is well organized and in concert with the figures in the paper looks to provide an excellent starting point for one wishing to replicate this technical note's

apparatus. With the exception of my minor specific comments below the design section of the manuscript clearly details the apparatus, its construction, and function of its components.

Section 2: Specific Comments:

1. Throughout the manuscript the authors use the term 'stable water isotopes' while this is a common shorthand, it is incorrect. There are no isotopes of water (molecules do not have isotopes), rather there are isotopes of atoms, or elements (hydrogen, oxygen, nitrogen and so on). The phrase 'stable isotopes of hydrogen and oxygen', is more precise (and correct). To quote Zachary Sharp's 2017 text Principles of Stable Isotope Geochemistry:

" Writing about 'water isotopes' may sound short and concise, but it is wrong. Just as petrologists don't talk about 'rock isotopes', so hydrologists should avoid talking about 'water isotopes'."

Table 2.1 in that text is an excellent resource for correcting often misused terminology. I suggest that the authors review that table and make corrections throughout their manuscript regarding the use of a number of terminological inconsistencies. Some of those issues are noted under this comment:

Introduction first sentence: Same stable water isotopes language issue referenced in comment 1, followed by the symbols for isotope ratios. It would be good to standardize the language throughout the manuscript in regard to isotopes vs isotope vs isotopic vs isotope ratios, and so on. From Zachary Sharps 2017 book:

"2.2.1 'Isotope' vs. 'Isotopic' These two words appear to be used randomly and interchangeably because the proper use of one or the other is not immediately clear. My mentor Jim O'Neil, was confronted with this dilemma as a U.S.G.S. employee. He consulted the Technical Reports Unit at the USGS for guidance. After some research, it was decided that 'isotope' is used when modified and 'isotopic' is used as a stand-alone adjective. One therefore should write "The oxygen isotope composition of . . ." and "The isotopic composition of .. .". In the first case, 'oxygen' modifies 'isotope' and in the second 'isotopic' stands alone"

If using the symbols for isotope ratios δ18O and δ2H, I would modify the first sentence to say: "Stable isotope ratios (δ18O and δ2H) have found…". Alternatively, one could say "Stable isotopes of hydrogen and oxygen (2H,1H, 18O, 16O) have found…", though I would lean towards the first.

Line 130-134: The authors use commonly applied shorthand's that are carry overs from oral communication: 'Heavy, and light water' in this case. I again refer the authours to Table 2.1 in Zachary Sharp's 2017 text in regard to more precise terminology:

" As numbers, δ -values can be high or low, positive or negative, but not heavy or light"(Sharp 2017 pg 2-3),

However, that said, I know what you mean when you say 'heavy or light isotope values', as do most other folks in our field, so I would leave the decision regarding this use of this terminology in the hands of the Editor, but I think we should strive to be as precise as possible with our language. I suggest the authors check for and address this issue throughout.

Line 165:Terminology issue: depleted signature and enriched signature, use precise terminology per above.

Thank you for recommending that book! We will change the wording to your and Sharps book suggestions.

We changed the wording.

2. Introduction paragraph 2 line ~30 "While most isotope analyzers…" I am confused by the meaning of this sentence. You seem to be saying two things here. First that isotope analyzers are not made for vapour analysis (?), and second that extraction of water from plants or soils is not possible (?)…but you then go on to describe existing extraction methods. Your first statement is not wholly correct as many isotopic analysis systems are built to analyse vapour or liquid samples. Some IRMS methods use vapour equilibration ($CO_2$ and $H_2$ equilibration) and both OA-ICOS and CRDS systems have vapour analysis modes (though not all of them do, especially older models). Are you trying to say that there is not an in-line isotope analysis systems which combines extraction and analysis? Because if that is the case, there is the Picaro induction module which is a peripheral system affixed to a CRDS analyzer that heats samples and inject the water vapour directly into the analyzer. I suggest this sentence be revised to improve clarity of meaning and to represent existing methodologies more accurately.

We were trying to say that extraction or equilibration is needed for soil- and plant samples. We will clarify this in the revision.

We restructured the sentence to clarify this.

3. Line 30 &31: Do you have a citation (or other data) which concretely indicates the popularity of CVD and DVE? And the rise in popularity of DVE?

We do not and therefore will exclude this sentence.

We excluded the popularity.

4. Line 35-47: This paragraph is leaning heavily on Grahler et al 2021. I suggest attributing the operating methodology described here to some of the original developers (Wassenaar, Hendry, Barbour, Pratt, etc). The use of silicone blot is an excellent idea though this is the first I've encountered it which is to say again a citation showing the introduction of this component would be useful (unless of course you are the progenitors of this technique). I have also seen a loop system used during DVE analysis where after inserting the needle and tubing to the sample bag for the inlet side connection to an OA-ICOS, a second connection is made with a needle and tubing connected to the analyzers outflow port (See Gaj et al 2019 DOI: 2136/vzj2018.04.0083). The latter technique allows for a continuous looping of the vapour sample through the analysis chamber and more stable readings.

We will give more credit to the original developers.

Regarding the silicone, this was also described by Grahler et al 2021.

The Idea with looping the air is a great idea for implementing the autosampler to OA-ICOS, however this would double the materials (like valves, valveblocks, etc.) needed. For CRDS, the air volume in the bag is totally sufficient to reach stability, therefore for now we will keep the setup as is.

We credited more authors of the original development.

5. Line 49-50: I don't find this statement to be entirely accurate. The Garvelmann study used 8 cm long soil cores (relatively large depending on water content) and the Wassenaar study used water volumes in both soil samples and liquid water samples around 20 ml. There are two main types of CVD. A larger glass and manifold style system like West et al 2006 (DOI: 10.1002/rcm.2456) or Orlowski et al 2013 (DOI:10.5194/jsss-2-179-2013), and a small capillary and vial systems described in Koeniger et al 2011(DOI: 10.1002/rcm.5198) and Millar et al 2018 and 2019 (DOI: 1002/rcm.8530). The Koeniger-style CVD system uses samples volume much smaller than would be required for DVE and in some cases depending on water content the sample volumes used in the larger West/Orlowski-style CVD system may be smaller than required for DVE. Sample size is contingent upon water content. Wassenaar 2008 discuss the required water content needed to limit any atmospheric effects. Hendry et al 2015 (Doi: DOI: 10.5194/hess-19-4427-2015) make mention of issues related to low water content. The Hendry study noting >5% GWC is needed for accurate measurement. In my experience with the DVE system, sample volumes need to be larger than I have used for both types of CVD method (again water content dependant). My main concern with the author's statement is that by saying "… small sample volume" they don't account for water volume requirements. I would change the language to exclude the use of 'small' and instead indicate that DVE can use sample volumes similar to those used in CVD depending on pre-extraction testing of water contents. This is important to not obfuscate that fact that in some cases relatively large sample volumes of soil or plant material could be required. The second part of this statement is accurate though: one of the great parts of DVE is that it can allow for high spatio-temporal resolution.

Thank you for the change suggestion. We will fix it, but also omit the paragraphs regarding CVD to focus more on DVE-LS.

We excluded CVD completely from the paper to focus more on DVE-LS, so we also deleted the comparison.

6. Line 55-56: The end of this sentence is not accurate. The research by Nehemy et al 2019 does not involve correction of isotopic data generated by DVE-LS as impacted by co-extracted VOCs. That research (and their follow up work in Millar et al 2021 (1002/rcm.9118)) were focused on developing a method for detecting spectral interference during DVE-LS analysis since a tool for detection of spectral contamination during vapour analyses for OA-ICOS did not yet exist. They do not suggest a correction approach using their detection method. Revisions here are needed

to correct for the inaccuracy of attributing the Nehemy et al reference with creating isotopic data correction techniques for spectrally contaminated samples.

We will change the sentence to more accurately represent the findings of Nehemy et al 2019.

We chose to exclude this paper.

 7. Line 59: Yes, I agree! The lack of standardization is a major issue across many aspects of ecohydrological study. The need for improvements in standardization of methods has been previously discussed in Ceperly et al 2024 (DOI: 10.1002/wat2.1727) and in Millar et al 2022 (DOI: 10.1002/hyp.14698).

Thank you!

We added citation to the two papers.

8. Line 70: Regarding the use of ambient air for flushing and memory effect. I see below that you used a drying canister during air flushing testing but found no improvements to your data. Our laboratory uses drierite canisters in between analysis of each sample to lower the analysis chamber water contents and to limit any issues related to memory effects. I am curious if during your testing you discovered any issues related to memory effects or if you did any testing to check for memory effects, given the use of ambient air for flushing? Did you run each sample more than once, or stager your samples so that waters with different isotope values were ran in an intermixed fashion? I would be concerned about any impacts from water in the ambient atmospheric air used during flushing given humidity levels can change drastically throughout a day, which may impact your measurements or contribute to some memory effects. I would also like to know if you tested for memory effects. The IAEA WICO survey from Wassesnaar et al 2021 (DOI: 10.1002/rcm.9193) indicated that some of the best performing laboratories were those that carried out multiple injections per sample keeping only the latter readings as a means of dealing with memory effect. The WICO tests were for liquid analyses, but the issue of memory effect is still relevant here, especially with such a high sample throughput!

We did not test for memory effect, but since the laboratory is climatized to a constant temperature, we do not expect larger intra-day humidity differences.

However to counteract memory effects, we did analyse the samples in a random sequence, with flushing periods in between and long analysis times until all our defined stability criteria were met.

Regarding the WICO tests, they were performed using liquid samples. With our DVE-LS bags, gas volume is limited, so therefore multiple measurements are not possible.

We added that the flushing removes possible memory effects

9. Line 84: 391 samples! Wow! An incredible improvement on daily sample throughput. This is great.

Thank you!

10. Line 106: Beginning "Within an active sequence…". I am slightly confused about the flushing sequence. If I am understanding this sentence and the preceding set up explanation, the CRDS is drawing air into its analysis chamber from the open ambient air flushing valve though the valve block to flush out any potential water, with the H2O value being measured in the CRDS analysis chamber yes? Or is there another senor in the valve block which measures the customizable H2O value? This needs clarification and is perhaps contradicting your discussion section 4.5 "Don't buy Cheap" which indicates that ambient air can cause measurement errors. Please clarify this. Second, when you say customizable H2O value, as user of such systems I am aware of this water values function (indicates the water content in the analysis chamber), but it may be worth adding a short sentence explaining what the H2O value is for non-expert users.

We will clarify that we only look at the H2O values given by the CRDS within its analysis chamber. We will also clarify the "Don't buy cheap" section; the issue is not about initially some mixing with ambient air, but about a continuous leaking-in.

We changed the wording to more accurately describe the procedure and changed the wording in the "Don't buy cheap" section, so it states the issue of the continuous mixing.

11. Line 127-128: How was airtight vapour flow ensured? If a specialized check was used here (beyond the use of the silicone blots), it bears mentioning so others may attempt to replicate this.

We did not implement any specialized check, we will clarify that this assumption is based on using proper materials which are individually rated airtight.

We changed the wording.

12. Line 170: As a counterpoint to the statement about accuracy vs measurement repeatability. Its bears noting that one can only present accuracy data if they have a know truth value to compare the subsequent measurements against. In cases where water to water extractions are carried out, truth values can be collected and accuracy assessed. However, if a study is using plant material, or soils, (and not carrying out a spiking experiment with all its well discussed issues Gaj et al 2017, Thielman 2019, Oerter et al 2014 etc. all reviewed in Millar et al 2022), a truth value may not exist and so repeatability is the only option is to present uncertainty and error. Wassenaar et al 2021 (Doi: 1002/rcm.9193) discuss the standards of error propagation calculations that we should all aspire to.

Knowing the "true value" is a real issue. In the revision we will go more into detail with measurement repeatability by applying standard deviation for the identical samples, as well as calculating the measurement bias (average difference of measurement to true value) for the whole system. This should produce a more robust statistical assessment.

We added measurement repeatability and bias to the assessment of accuracy

13. Line 176: on your new systems accuracy: Your accuracies fit within the ranges described for other DVE publications in Table 2 in Millar et al 2022 which looks to have added new publications for vegetation extraction and new methods from what was first shown in Springer et al 2015.

Allen and Kirchner 2022 (DOI: 10.1002/hyp.14483) discuss the issue of accuracy/error and range of end members, while lower accuracy is not always desirable, high throughput has a benefit and, in some cases, depending on end member range it may not be a huge issue if the error is 7‰ for δ2H for example.

Yes, we think higher throughput can sometimes also be a worthy trade-off to accuracy,a as long as it is within acceptable limits.

We added a paragraph discussing the implications of end member ranges and the applicability of the VapAuSa

14. Line 180: The Orlowski study used zip-top plastic bags as opposed to the new standards set by Grahler: (aluminum laminated with plastic coffee-style bags) and discussed through a lens of diffusion rates in Millar et al 2022). I expect the cost of these aluminum bags ,while higher than standard market available resealable zip-top bags, are still lower than the cost of the consumable glassware, and liquid nitrogen utilized in the CVD approach. Are there any newer citations which also discuss cost and time of set up given Grahler's findings on appropriate bags? If no citation exists can the authors discuss briefly a cost comparison from their own use of such systems?

We will try to give a cost comparison in the revision. Compared to the labour cost of manual analysis, the consumable-material cost is nearly negligible for DVE-LS.

We chose to exclude the comparison to CVD as the paper should fovus on DVE-LS, therefore we will not include a cost comparison

15. Line 183: The automation of this process will be a boon to users of the DVE-CRDS system! If only it existed for OA-ICOS as well.

The project is open-source, maybe someone can adapt it!

16. Comments for Section 4.3 on storage time:

Line 186:  "Uncertainty might cancel out due to lower sample storage time". This kind of speculation is not useful without data to back up it up. It is hopeful to say that may be the case but as scientists we do not rely on might or could be. I would revise this sentence or provide data/ citations which support it.

We will include a comparison of storage induced and autosampler induced uncertainty.

We changed the wording to better explain our concept of
*less storage time = lower uncertainty* and *VapAuSa = less storage time* therefore *VapAuSa = lower storage induced uncertainty*

Line 187: "Over time these changes appear with every…" Do you have a citation for this? Such a statement requires one. I know that some users heat seal their Al-Plastic bags as opposed to using the zip-top. Is there any data on leaks from heat sealed bags?

We gave a citation to that statement, the drifts can be found in Gralher et al., 2021. They also compared heat sealed and only zip-locked bags. We will add some more information here.

Issue continued in line 190: Again scientist should not assume, they should provide data or citations which confirm their reasoning. Research has been carried out assessing temperature effects on isotopic drift. Millar et al 2022 thoroughly review storage material effects and temperature effect of the isotope values of stored samples. I suggest the authors review the references detailed in that work instead of making assumptions regarding freezer storage as some research had already been done regarding this issue.

Thank you for the paper suggestion, we will incorporate it.

17. Section 4.4 On objectivity. This is very good. The field need more transparent reporting of all parameters applied during extractions. Extraction parameters should be reported in all publications using them.

Thank you!

18. Line 200: The statement regarding operator body heat seems speculative, but I could be wrong. Do the authors have data or citations to support this?

We will clarify this in the revision.

We chose to exclude this statement as the effect we see when measuring manual vs. by Autosampler as we can not exclude other factors beside operator body heat

19. Section 4.5 Don't buy cheap: Mentions that even a small amount of ambient air leaking into the system can cause errors dung vapour analysis. But this contradicts the use of ambient air to flush their system. Given my comments above regarding clarification of the ambient air flushing system, if ambient air is not in fact pulled into the measurement chamber during flushing, then this is not a contradiction. Clarification is needed above so the readers are fully certain that ambient air is not being pulled into the measurement cavity during flushing of the block. However, if ambient air is pulled in the measurement chamber during flushing, then this section is contradictory to the above design. Please clarify.

As stated above, we will change the wording and explain the differences between permanent leakage and flushing.

We changed the wording to the issue of continuous mixing.

20. Line 217: The authors should not suggest use of cheap plastic bags in their conclusion. Grahler's research on appropriate materials for sample storage and analysis during DVE-LS seems conclusive as does the discussions in Millar et al 2022 indicating that zip-top plastic bags are not appropriate for use with DVE-LS. The addition of this statement in the conclusion does not take into account the fact that some samples collected in zip-top bags may undergo periods of storage or require transport (time) which increase evaporative fractionation risks. Introducing this idea in the conclusion is not appropriate but should they wish to make this point it should

be expanded upon in the discussion and the leak risk compared against other already published studies which indicate these types of bags are not appropriate.

We will discard this point.

We discarded it.

Section 3: Technical Corrections:

1. Language correction: In abstract ~ line 5: "However, sample analysis requires a significant manual labor, thereby limiting the number of samples that can be analyzed.". Remove the 'a' between 'requires' and 'significant'.

We will change this.

We changed it.

2. Language correction: Introduction paragraph 2 line ~30 "While most isotope analyzers are build…" should be 'built' not 'build'.

We will change this.

We restructured the whole sentence.

3. Introduction paragraph 2. I think the structure of this paragraph could be reworked to improve readability. You begin with a confusing statement which is not accurate depending on your meaning (discussed in my comments above), briefly mention DVE, then immediately change topic to discussing CVD and how it works. The paragraph ends and the next takes up the discussion of DVE. I understand that you may want to mention the popularity of DVE and CVD together, but I find this disjointed.

We will discard CVD from the introduction.

We discarded CVD from the manuscript

4. Line 49: "Additionally on a small sample…" space required between volume and similar

We will change this.

We changed this

5. Line 30-31: DVE-LS is only partially defined. I know that LS = laser spectroscopy but other readers may not, so the acronym needs to be fully defined.

We will clarify the abbrevation.

We clarified this

6. Line 74: Language issue for sentence: "For 23 sample  bags are connected with cannulas to the valves through 1/8 inch PTFE tubing". The use of 'For' at the start of sentence is incorrect or something is missing later in the sentence. Depending on desired meaning 'For' could be removed, simply saying "23 sample bags can be connected via cannulae to…"

We will change this.

We changed this.

**Authors' responses to the comments of reviewer#2**

Thank you for reviewing and commenting on our manuscript, "Technical note: A fast and objective autosampler for direct vapor equilibration isotope measurements". We find your advice constructive and will incorporate your suggestions into our revision. We responded to each comment individually below.

Thank you for letting me review this manuscript. The authors present an autosampler for measuring the water isotope composition of water (and potentially, soil and plant) samples using the direct vapor equilibration method. I know and appreciate the work of the group and I am user, hence, very familiar with the DVE-LS method as well. Providing an open-source autosampler for increasing sample throughput and minimizing labor and cost is simply amazing and I applaud the others for developing and providing all codes publicly. However - and I am sorry to say this - the presented study has severe inconsistencies and shortcomings, which simply need to be addressed before publishing. In brief, I could summarize these as follows: i.) for a convincing proof-of-concept, the results for measuring 21 samples is simply insufficient; ii.), the ms. refers to soil samples, CVD and artifacts throughout, but does not actually report any results for using the autosampler on soil sample equilibrations, which would be much more interesting and relevant for the actual application of the method; and iii.) potentially relevant factors were not tested for nor commented on (e.g., the effect of leaving the samples to be measured pinched by the canules). Additionally, the structure of the ms is currently confusing and is missing a red line (mainly due to ii.) I believe). I hope my comments and suggestions help in improving the ms. Having that said, I totally support publication of the autosampler, after the data foundation for a true objective evaluation is improved.

General/Main comments:

   Were only 21 samples measured in total? This is very low for assessing accuracy/uncertainty and makes me seriously wonder – the authors have the autosampler, why would they rely on such a small number of samples for proofing its accuracy and uncertainties?

We have already measured 1000+ soil samples with the VapAuSa. However we did not report on these samples as we did not co-measure them by hand. These samples originated from mountainous catchment soil profiles where the seasonality- and altitude effect of precipitation is difficult to assess. Therefore, assessing the measurement uncertainty would be nearly impossible.

However, to address this and the following point, we will take more soil samples and co-measure them by hand and autosampler and report on the findings in the next manuscript revision.

We took 90 additional soil samples which we co-measured by hand and VapAuSa. As we measured each bag twice, we measured some by hand first and some by autosampler first. Tjis is all added to the revised manuscript. Since we do not know the "true" liquid values of the soil samples, an assessment of measurement repeatability and bias is still necessary with liquid water samples.

The manuscript mentions soil samples, CVD, and artifacts introduced by extraction multiple times and quite prominent; however, no soil data is presented in the manuscript at all. This is confusing me extremely. Firstly, CVD is not of relevance for the autosampler at all for the purpose of proof-of-concept. Secondly, I would've loved to see comparisons between manual injections and autosampler results for soil samples! I don't understand why this is not part of the study.

As stated before, we will take soil samples and measure them manually and by autosampler.

As stated above, this was added as sections in the method, results and discussion sections

155-165: This chapter is completely unrelated to the actual study, which was done using liquid water, not soil samples. Also, the 'bashing' of CVD and explanation of the differences or not of any importance for the presented study. I strongly suggest to focus on comparing and testing the accuracy and uncertainty of the data obtained with the autosampler vs. manual injection and liquid water measurements for comparable matrices, i.e., liquid, non-extracted water.

We will discard the whole section regarding CVD and focus on the DVE-LS method improvements by the autosampler.

We excluded the comparison of CVD and DVE-LS from the manuscript.

Same as above is true for chapter 4.2: The study did not compare data obtain from CVD with data obtained by DVE-LS – why is this discussed in the manuscript? I suggest again to focus on the actual purpose of the study: Provide an autosampler for water vapor isotope measurements and proof its accuracy, benefits and caveats. The methodological artifacts introduced by such a study must be eliminated, i.e., by only using water samples.

We think this point is already addressed by our water-to-water extraction of DVE-LS, done manually, by autosampler as well as liquid measurement. However we will try to improve the clarity of the section so the suggested points are addressed more clearly.

We changed the focus to your suggestion.

Avoid colloquial language and vague statements ("some", "lots of", e.g., l.195)

We will rephrase the content to scientific language.

We changed the wording and gave exact statements

Minor comments:

Abstract:

Vapor or vapour? à use consistent BE or AE

Thank you for pointing this out, we will revise it to the AE version

We changed all occurrences to the AE version

12: the reported uncertainty of both manual and automated measurement for d2H seems pretty high, compared to liquid injections. It would be great to see some elaborations on the reasons and/or recommendations to improve this somewhere in the ms.

Yes, the uncertainties are pretty high, we will address this in the revised manuscript

We added a paragraph stating the applicability of VapAuSa depending on the delta value range of the end members studied. If the range is small, hand sampling should be preferred. But if the range is substantially larger than the added uncertainty, then the benefits (time & throughput) of VapAuSa overwheigh.

Please report here on the methods also (how was the reported accuracy etc. elaborated – how many samples of which matrix were tested)

Regarding this and the following point: We will rework the accuracy section to incorporate measurement repeatability (as standard deviation) as well as giving measurement bias (average deviation from measurement to "true" liquid measurement). This will then also be part of the abstract.

Why is the later mentioned accuracy not reported in the abstract? (and also, repeatability)

We changed the abstract to incorporate this and the previous point.

Text:

20: obtain estimate

We will change this in the revision

We changed the wording

24: distribution composition

We will make this alteration in the manuscript

We changed the wording

26: partitioning effects à unclear what is meant with this, please rephrase or explain. Suggestion: water partitioning

Thank you for the suggestion, we will change it

We incorporated your wording

29/30: this sentence is unclear, it is very much possible to extract water from soils and plants; isotope analyzers are also not built to measure specifically liquid water; in fact, the lasers all measure water vapor. The autosamplers though are build for liquid water, I guess this is what is meant here. Please rephrase.

Yes, we wanted to state that the isotope analyzers (CRDS and OA-ICOS) will need water-vapor as an input to measure. Therefore an extraction and vaporization or an equilibration is necessary. We will rephrase the content for more clarity.

We have rewritten the sentence, also changing the following point.

> 31: DVE is not really an extraction, it's an equilibration

We will clarify this.

> 32-34: the explanation of CVD is explained very sloppy; from the current explanation it is absolutely not clear how CVD works. Also, the citation used is very old – the method has been much improved meanwhile (classic citation Koeniger et al., 2011, but also many others)

As stated above, we will discard the whole section regarding CVD and focus on the DVE-LS method improvements by the autosampler.

We excluded CVD from the manuscript to focus on the improvement of DVE-LS

> 38: For other users, a reference to the company and size of the laminated bags would be useful here

Thank you for pointing this out, we will name the manufacturer and sizes used in the application

We added the manufacturer and bag type.

> 49: volume similar (space missing)

We will change this

We changed it.

> 49/50: CVD requires much less soil compared to DVE-LS, only 10g are sufficient for one replicate in many instances. Having that said, it should be stated somewhere how much soil is needed. Also, the constraint for high spatial and temporal sampling is often the digging, not the sample storage or amount.

We will change this paragraph to more clearly state the requirements and discard the CVD comparison. In our experience, the drilling was not as limiting as the sample processing and analysis. However we acknowledge that both can be limiting depending on your available field- and laboratory-staff and equipment. Therefore we will rephrase this statement.

We changed the paragraph to only focus on DVE-LS, therefore excluding this part.

> 74: please provide details of the cannulas (diameter) – it might be a constraint for providing the 35ml/l to the Picarro and induce errors if diameter is too small

Thank you for pointing this important fact out! We will state the diameter (2.1 mm) in the next revision

We added the diameter and length as well as manufacturer and product name

127: It seems like the cannulas all have to be connected to the bags in the beginning of the measurement cycle. This means that the later they are in the sampling line, the more potential exists for contaminating the sample because the sample was pinched. It would be great to have some comments on this. Also, silicone might affect the absorption spectra of the measurements – could this be explanations for the relatively high uncertainties of the method? It would be nice to see some elaborations on this.

We will add this to the manuscript. So far what the data of the 21 bags tell us is that there is no effect on later samples by the pinching. However we only sampled for 5 hours. With larger systems changes might get more pronounced. Therefore we will also do a longer test and report on it in the revision

Regarding the silicone, we think this effect is minimal as other studies which also applied the silicone had lower uncertainties (e.g. Gralher et al., 2021)

We added a paragraph in the discussion where the pinching time and drift is examined.

129: Just referring to the table is not sufficient, please explain what the stability criteria actually checks.

We will improve this in the revised manuscript

We added an explanation in the text.

135: next to uncertainty, the accuracy of the method should also be reported in the abstract

We will add this.

We chose to recalculate the accuracies as measurement repeatability (1 standard deviation) as well as measurement bias (average difference of measurement to true value)

152/153: Perhaps the authors are right, but this needs to be further discussed. As stated before, there are potential influencing factors which could cause a difference of the manual vs. automated injections; e.g., the time until a sampling is analyzed (intrusion of atmospheric air?) and the potential contamination with silicone (could affect only one isotope and hence, explain different behavior of d2H and d18O).

As stated before, we will assess the effect of time until sampling and don't think silicone will be such a contaminating source (after all, the Picarro liquid vial caps are also lined with silicone). We will further discuss this uncertainty in the revision.

We excluded the last sentence as it is more a discussion than a result. As stated above, we added a section discussing the influence of pinching the bags and later measurement.

173/174: I agree that ONLY reporting repeatability is masking the actual outcomes, but not checking for repeatability is equally questionable when providing a proof-of-concept. Why not report both?

As stated above, we will rework the accuracy section to incorporate repeatability as well as measurement bias. The measurement bias is <0.1e-13 for both isotopes, therefore we did not

think about reporting it. But we agree that giving this might be beneficial, therefore we will add it to the revised manuscript

As stated above, we added repeatability and measurement bias to the evaluation.

General suggestion for the discussion: Try to stay objective in the statements. Reading the discussion, I felt that the authors are trying to convince me how good the proposed method is, rather than objectively reporting and discussing the pros and cons.

We will rephrase the content to be more objective.

By excluding CVD from the paper, we hope the discussion is now objective and solely focuses on the improvement of DVE-LS

4.3: The elaborations here are good, but please comment on the potential effect of pinched bags during the measurement cycle

We will add this to the manuscript.

We added it.

216: may be is ;)

Thank you, we will rephrase it!

We chose to keep it.

219/220: "So far we tested the system with liquid and soil samples only" Why are these results not presented here?

As stated, wee will add this in the manuscript

We have rewritten the conclusion to incorporate the soil samples.

**Authors' responses to the comments of reviewer#3, Fabio Marzaioli**

Thank you for reviewing and commenting on our manuscript "Technical note: A fast and objective autosampler for direct vapor equilibration isotope measurements". We found the advice constructive and will incorporated the suggestions into our revision. We've responded to each comment individually below.

The authors have developed an automated system, called VapAuSa, for water isotope measurements by direct vapour equilibration (DVE-LS). The system is described with a lot of detail, but the data reporting and commenting is still poor in my opinion. I see a strong problem in the definition of accuracy and the results obtained are hardly commentable in this sense. The data are there, but some work should be done to produce scientifically meaningful results.

Detailed comments.

Abstract

Ln 7: What do the authors mean for lack of objectivity?

In regular direct-vapor-equillibration laser-spectroscopy (DVE-LS) measurements, the measurements fluctuate during anlysis. The operator has to subjectively decide when the measurements look stable, and then note the analyzer readings.

We clarified what we mean with the lack of objectivity in the Abstract.

Ln 12: Can you please specify what is your precision estimator? Are they comparable according to a Fisher test?

We will change the accuracy metrics to repeatability (applying standard deviation) and measurement bias (average difference of measurement to true value).

We added measurement repeatability (1 standard deviation) and measurement bias (average difference of measurement to true value) to the paper

Intro

Ln 29: I would suggest not using "isotope analysers". It goes in the verse of a measurement black box while if one knows principles of measurements for apparatuses can be a great help to produce accurate data.

We will avoid using "isotope analyzers" and change it to the accurate device description (cavity ring-down spectroscopy (CRDS) or laser spectrometer)

We changed the senctence.

Ln 29: Please avoid the usage of isotopic composition (refer to isotope ratio od delta).

We will change the wording to the scientifically correct terms.

We changed composition to ratio throughout the paper

Ln 32: Please rephrase, how does a vial contain vacuum? The whole process happens under vacuum.

We will discard the whole part on cryogenic extraction to focus more on direct-vapor-equillibration.

We excluded CVD from the paper

Ln 37: add Gas before tight.

We will adapt the wording.

We changed it to "gas diffusion tight"

Ln 38: I would not see the 2) as a different point form the 1).

We will summarize both under point 1)

We summarized the points

Ln 42: Isotopically Equilibrate: change to "can reach isotope equilibrium".

We will change the wording to your suggestion.

We changed this to your suggestion

Ln 49: volumesimilar: volume similar?

We will correct that.

We corrected it

Par 3.1

Ln 123: 3 not three.

According to the HESS style guideline we will leave it at three ("For items other than units of time or measure, use words for cardinal numbers less than 10; use numerals for 10 and above (e.g. three flasks, seven trees, 6 m, 9 d, 10 desks)"; https://www.hydrology-and-earth-system-sciences.net/submission.html#english).

We kept it as "three" due to the style guidelines

Table 1: Tables can not be used without IS units.

Good point, we will fix that.

We added SI units

Period starting at line 130: There should be uncertainties for the reported water samples. Only primary standards are reported with no uncertainties.

Thank you for pointing this out; the uncertainties are only shown and stated in figure 5. We will ad it to the text.

We added the measurement uncertainties of the standard waters

Ln 135 and soon after: Accuracy cannot be a difference between 2 values. Of course, since a and l are measured, they are affected by uncertainties. Therefore, an unbiased way to define accuracy is to define a variable whose numerator is the reported difference (accuracy d18O formula) and whose denominator is the propagated uncertainty of such a difference.

We now calculated measurement repeatability by applying standard deviation for the identical samples, as well as calculating the measurement bias (average difference of measurement to true value) for the whole system. This should produce a more robust statistical assessment.

As stated above, we changed the accuracy section to incorporate the measurement repeatability and measurement bias and changed the formula name to "deviation" # MUSS NOCH ANDERS

Par 4.1 (Accuracy)

The section is not homogeneous. The introductory part should be included in the intro. Also, if some authors (ln 170) used repeatability as an indicator of accuracy, it should be stated that this is a misinterpretation. Also, it is not a matter of ranking accuracy vs repeatability, they are 2 different concepts.

We will introduce the section already in the intro. Yes, no uncertainty metric is inferior to the other, however we think some are better suited to assess the performance of a new device than others. Measurements can be highly inaccurate even when repeatability is high. Ideally, authors should state both metrics.

---

## Referee Report (RR1)

**Review of Revision Technical note: A fast and objective autosampler for direct vapor equilibration isotope measurements.**

Jonas Pyschik et al.

**Overall**

This paper introduces "VapAuSa," (not sure how this is pronounced) a new automated vapor-equilibration autosampler system designed for implementing the direct vapor equilibration method for H and O isotope measurements of soil porewater (or any medium with water in it). It advances isotope analyses of porewater by improving sample throughput, reducing manual labor, and improving the reliability and reproducibility of stable isotope analysis, particularly for soil porewater samples. Previously, the approach was manual, laborious, involved many steps, and was unique to operational procedures of each laboratory, which often led to systematic difference between laboratories. The benefits of automation are a significant increase in sample throughput, a 90 % reduction of labor, and generally standardizing the measurement process, thereby reducing potential for human error, and minimizing potential evaporitic isotopic changes in stored samples due to faster processing times. This is an invaluable advance for porewater isotopic measurements, and hopefully can achieve a wider adoption by some form of commercialization.

**Recommendation**: Accept, with major revisions. Some key critical information is missing that must be added (see Major comments).

**Writing Style**

Regarding scientific writing style, there is widespread inconsistency in the manuscript in grammatical tense, causing confusion. The paper needs to be edited to correct this. It seems a minor thing, but it will improve how the paper reads. Here is the rule to follow: YOUR work and findings must be stated in the past tense (did, determined, tested, had, were), but all published citation findings and well-known knowledge are given in present tense (is, are). An example of this inconsistency referring to the same published work (present tense) is in Lines 19 "they are applied…" and Line 21 "they  ARE also used to estimate..". Many other instances.

**Major Comments**

**Hardware** – how are samples thermally controlled? You need to add a descriptive paragraph about this because it a critical aspect to back determine the porewater delta values. Are sample boxes held in an insulated box, T-controlled room? How is T controlled and to what precision. Etc. What is recommended for users?

**Sample bags** – give a description (Part No., volume etc.) in the Methods and a clear summary of the procedure how samples are added and sealed – this is missing (but briefly referred to later). At least one detailed paragraph regarding this aspect of the operation is needed. This must be linked to software timing (a small bag will deflate quicky etc.). Note too the gas sampling flow rate from a Picarro (CRDS) much slower than an LGR (ICOS) that is mentioned as compatible with the system.

**Minor Comments**

Title: The word "objective" is not appropriate, and you are missing the main object of the assays (porewater). Suggest changing to "An autosampler for rapid and reproducible direct-vapor-equilibration stable isotope measurements ($\delta^2$H, $\delta^{18}$O) of porewater"

Line 3. "...bonded to substances like...". Bonded is a poor work choice here (suggests a chemical structure). Suggest changing to "... water in or adsorbed to..."

Line 6. "... may undergo evaporative isotopic changes... "

Line 7. "lack objectivity" – again poor work choice. Suggest "...manual measurements require many laborious procedural steps that can easily compromise reproducibility."

Line 7 – delete "currently"

Line 10 - "...connect to a laser isotope analyzer... automated measurements."

Line 11 "... performance criteria can be specified... facilitating reproducible analyses"

Line 27 (and other locations) – I realize this can "theoretically" be used for plant water, however, its well known that plant water has a lot of VOCs that wreak havoc with laser analyses through spectral interference. In the experience of this reviewer, DVE does not work well on plant water and even on plant water extracts, unless there is cryogenic purification. Because plants are not the focus of this work, I strongly suggest leaving out any references to plant water unless you can prove with data that it works.

Lines 39-42 – Somewhere in this section you need to emphasize that thermal stability is absolutely critical (and tolerance) because you are using the highly temperature dependent isotope fractionation between measured vapor and the liquid to back calculate the porewater isotopic composition. Also give the equilibrium equation used, e.g. Majoube and others ...

Line 55 alters through? .... evaporation and diffusion...

Global: never use the term isotopic "signature" – use "values" instead.

Line 56 replace "objectivity" with "automation" (everywhere in the paper).

Line 57 – stray ? in Ceperley reference

Line 59 delete "autonomous" use "automated"

Software – are there any failure monitoring features – leak, pumped the bag too long etc.?

Line 123 – how much soil place in the 500 mL bags? Is this ratio important?

Line 125 – samples were stored at 20 C ± 1 C – were samples measured in the same room, and was this potential error in T factored into the reported analytical uncertainty?

Line 163 - should be superscript on delta values

Line 169 – "pinched" – do you mean "punctured" ? - unclear.

Line 175-176 - ??? means differences were -XX e-15???

Line 221 – you need to explain the reasons for drift (only evaporation?) and how it can be combated.

4.3 Objectivity (not a good word)  -> Improving Replication?

Line 248  - replace personal with manual

Line 251 – avoid speculating about plant water as noted.

---

## Author Response (AR2)

**Report #1**

We appreciate your review and comments on our manuscript, "Technical note: A fast and objective autosampler for direct vapor equilibration isotope measurements". Your feedback is valuable to us and we will make the recommended minor revisions accordingly. However, we do not agree with the major revisions you ask for, as all the points raised are already addressed in the manuscript. We provide detailed responses to each of your comments below.

Writing Style

Regarding scientific writing style, there is widespread inconsistency in the manuscript in grammatical tense, causing confusion. The paper needs to be edited to correct this. It seems a minor thing, but it will improve how the paper reads. Here is the rule to follow: YOUR work and findings must be stated in the past tense (did, determined, tested, had, were), but all published citation findings and well-known knowledge are given in present tense (is, are). An example of this inconsistency referring to the same published work (present tense) is in Lines 19 "they are applied…" and Line 21 "they were ARE also used to estimate..". Many other instances.

We partly disagree with this comment. While we agree that our work and findings must be presented in the past tense, we use the present tense when we directly refer to figure to explain them and for continuously relevant published work (like in line 19). However, when the publication is in the past, but still relevant, we use the present perfect tense (like line 21). We edit the paper again according these general guidelines.

Major Comments

Hardware – how are samples thermally controlled? You need to add a descriptive paragraph about this because it a critical aspect to back determine the porewater delta values. Are sample boxes held in an insulated box, T-controlled room? How is T controlled and to what precision. Etc. What is recommended for users?

This is stated in Line 125: "The prepared bags were then stored in a climate-controlled room maintained at 20°C ± 1°C for 48 hours." We added a sentence that the room in which they were stored is the same as where the are measured, so it is clearer

Sample bags – give a description (Part No., volume etc.) in the Methods and a clear summary of the procedure how samples are added and sealed – this is missing (but briefly referred to later). At least one detailed paragraph regarding this aspect of the operation is needed.

Again, this was already addressed in Line 123, and exactly in the format that you want us to revise it to: "These samples were then placed into aluminium-laminated plastic bags (WEBER Packing GmbH; CB400-420BRZ; 500 ml) and initially sealed with a ziplock. Afterwards, the bags were inflated with dry air, heat-sealed, and equipped with two silicone blots to ensure each measurement started with a "fresh" septum "

This must be linked to software timing (a small bag will deflate quicky etc.). Note too the gas sampling flow rate from a Picarro (CRDS) much slower than an LGR (ICOS) that is mentioned as compatible with the system.

We never mentioned that the system as it is is directly  compatible to a LGR (OA-ICOS). Everything is made for Picarros and consistently stated so in the manuscript!

Minor Comments

Title: The word "objective" is not appropriate, and you are missing the main object of the assays (porewater). Suggest changing to "An autosampler for rapid and reproducible direct-vapor-equilibration stable isotope measurements ($\delta^2H$, $\delta^{18}O$) of porewater"

We like the suggested word "reproducible" and changed the wording in the manuscript and the title.

Line 3. "...bonded to substances like...". Bonded is a poor work choice here (suggests a chemical structure). Suggest changing to "… water in or adsorbed to…"

We changed that.

Line 6. "… may undergo evaporative isotopic changes… "

We added this.

Line 7. "lack objectivity" – again poor work choice. Suggest "…manual measurements require many laborious procedural steps that can easily compromise reproducibility."

As stated above, we changed it throughout the manuscript.

Line 7 – delete "currently"

We deleted it.

Line 10 - "…connect to a laser isotope analyzer… automated measurements."

We added this.

Line 11 "… performance criteria can be specified… facilitating reproducible analyses"

We changed the wording.

Line 27 (and other locations) – I realize this can "theoretically" be used for plant water, however, its well known that plant water has a lot of VOCs that wreak havoc with laser analyses through spectral interference. In the experience of this reviewer, DVE does not work well on plant water and even on plant water extracts, unless there is cryogenic purification. Because plants are not the focus of this work, I strongly suggest leaving out any references to plant water unless you can prove with data that it works.

The decision to apply DVE-LS ultimately rests with the researcher, who must determine whether it is a suitable method for their study. In this context, VapAuSa is just an automation for DVE-LS. The cited studies in Line 27 by Bertrand et al. (2014) (https://onlinelibrary.wiley.com/doi/abs/10.1002/eco.1347), Smith et al. (2020) (https://onlinelibrary.wiley.com/doi/abs/10.1002/hyp.13838), and Kuhlemann et al. (2020) (https://doi.org/10.5194/hess-2020-425), all employed DVE-LS successfully for plant samples.

While we acknowledged the co-extraction of VOCs as a potential issue, this influence can be corrected, as demonstrated by Hendry et al. (2011) (https://doi.org/10.1021/ac201341p) and recently also by Hebstritt et al. (2024) (https://doi.org/10.1002/rcm.9907). Moreover, Millar et al. (2018) (https://onlinelibrary.wiley.com/doi/abs/10.1002/rcm.8136) found that for plant samples, "direct

vapor equilibration outperformed all other methods" due to reduced co-extraction of volatile organic compounds. For these reasons, we have decided to retain this method in the manuscript.

Lines 39-42 – Somewhere in this section you need to emphasize that thermal stability is absolutely critical (and tolerance) because you are using the highly temperature dependent isotope fractionation between measured vapor and the liquid to back calculate the porewater isotopic composition. Also give the equilibrium equation used, e.g. Majoube and others …

We added a sentence to the relevance of temperature stability.

Line 55 alters through? …. evaporation and diffusion…

We added " evaporation and diffusion" for clarification.

Global: never use the term isotopic "signature" – use "values" instead.

We changed it to "composition" or "value" throughout the manuscript.

Line 56 replace "objectivity" with "automation" (everywhere in the paper).

We replaced it with "reproducibility".

Line 57 – stray ? in Ceperley reference

We fixed the citation.

Line 59 delete "autonomous" use "automated"

We changed that.

Software – are there any failure monitoring features – leak, pumped the bag too long etc.?

No, there are currently not.

Line 123 – how much soil place in the 500 mL bags? Is this ratio important?

We added the amount (2-4 Tablespoons). The ratio is less important as long as there is enough water in the soil.

Line 125 – samples were stored at 20 C ± 1 C – were samples measured in the same room, and was this potential error in T factored into the reported analytical uncertainty?

The samples were measured in the same room. The potential error of T was not factored in, as T was not recorded. However, all samples were ppm ($H_2O$) corrected, which resolves temperature changes. We added this to the manuscript.

Line 163 - should be superscript on delta values

We updated the formatting.

Line 169 – "pinched" – do you mean "punctured" ? - unclear.

We changed the wording.

Line 175-176 - ??? means differences were -XX e-15???

The bias means were that low, not the differences. So there is no bias.

Line 221 – you need to explain the reasons for drift (only evaporation?) and how it can be combated.

We added evaporation and diffusion as drift reasons. Also we added that they should be cooled

4.3 Objectivity (not a good word) -> Improving Replication?

We changed it to "reproducibility".

Line 248 - replace personal with manual

We changed this.

Line 251 – avoid speculating about plant water as noted.

As stated above, we will keep it in.

**Report #2**

We appreciate your review and comments on our manuscript, "Technical note: A fast and objective autosampler for direct vapor equilibration isotope measurements". Your feedback is valuable to us, and we will make the recommended revisions accordingly. We provide detailed responses to each of your comments below.

The manuscript: "Technical note: A fast and objective autosampler for direct vapor equilibration isotope measurements" (egusphere-2024-528) Submitted on 22 Feb 2024 by Jonas Pyschik, Stefan Seeger, Barbara Herbstritt, and Markus Weiler has been drastically improved after (1st round) reviewing. There are still some important pitfalls mostly regarding the presentation of results. Here by I "paste" a list of detailed suggestions: "

Ln 1: stable water isotopes to stable isotopes (2H and 18O) ratios of water

We changed this.

Ln 14: 0.1e-12 = 1e-13 ‰ (means 1e-16 on absolute scale?)

We changed the value.

Ln 15: Have you performed a F test? Please report values of significance

We did not perform an F-Test but calculated measurement repeatability by applying standard deviation for the identical samples and calculated the measurement bias as the average difference of measurement to true value for the whole system. So there are no p-Values to report.

Ln 19: add abundances after isotopes

We like the wording of how it is currently written ad will leave it this way.

Ln 20: why not hydrogeological?

We added that.

Ln 23: if you need biblio please evaluate to cite: "Nasta, P., Todini-Zicavo, D., Zuecco, G., Marchina, C., Penna, D., McDonnell, J. J., … Romano, N. (2023). Quantifying irrigation uptake in olive trees: a proof-of-concept approach combining isotope tracing and Hydrus-1D. Hydrological Sciences Journal, 68(10), 1479–1486. https://doi.org/10.1080/02626667.2023.2218552"

In the intro, we only cite papers applying DVE-LS. Since the suggested paper applies CVD we will exclude it.

Ln50: Instead of "referenced" I would use "normalized".

We changed the wording.

Ln 130: Performed instead of conducted

We changed that.

Table 1. add [units] after sd, sd has the same dimensions and measurement units of physical quantities.

We changed it to your suggestion.

Ln 159-60-61: It is not correct presenting errors with such a number of digits! Only 1 or 2 significative digits are allowed and reported values has to be cut appropriately. E.g. ln 159: 0.7 (+/-) 0.035 wrong! You must write 0.7x (+/-) 0.04 or 0.7xx (+/-) 0.035. Once again I regret that in a scientific paper this rule falls to be applied.

Thank you for pointing this out, we corrected it to a x.xx digit format.

Ln 184: sd with only 1 or maximum 2 digits

We also changed this to two digits.

Ln 181: "Although the standard deviation was relatively high, it is similar across the measurements and suggests that the variance is likely due to the bags being measured twice, a theme that will later be discussed." Does this sentence refers to the secondly measured bags only? Why do no show box plot also for these sets?

It refers to the fact that we measured each bag once manually and then by autosampler or vice versa. This is what is displayed in Fig. 5, where we show all data.

Figure 5 GENERAL IDEA: Why do not you comment that both datasets plotted intercept zero in the text?

We added this to the text.

Ln185: If you refer to r^2 which should be written in capital letter R^2 this is the squared correlation coefficient. It is used to evaluate linear relationship between two variables. Moreover you should explicitly state what are the two variable that you want to correlate linearly.

We changed this and added upon your suggestion.

GENERAL IDEA
Once you applied such linear model to produce figure 6 why do not evaluate and comment slope values and intercepts? Theoretically one should expect that such relationship with no biases should produce a 1:1 line with 0 intercept. In such a case it is better to add to the plot 1:1 line and also use the same scales for x and y.

The line shown in Figure 6 is the 1:1 relationship, which we believe shows the divergence. However, we added to the figure caption that the line is the 1:1 ratio, which indicates the optimal ratio.

Ln 197 e 198 reported values are much smaller than what reported in the abs.

The measurement repeatability is identical to that in the abstract. The low values are the measurement bias and the individual water samples.

Ln 198: sigma for dD can be cut to 4 per mil

We changed that.

FIGURE 7 please try to improve figure 5. My suggestion is to ad data to figure 5 to produce something easily ascribable to FIGURE 7. Moreover why there are dD missing in figure 5?

We now also plot dD in Fig. 5 & 6.

Ln 230: accuracies also should be reported with the same number of digits.
i.e. 0.7-1.0 no 0.7-1.03.

We changed this.

Ln 240: source of uncertainty not uncertainty variable

We changed this.

---

## Author Response (AR3)

**Report #1**

We appreciate your review and comments on our manuscript, "Technical note: A fast and reproducible autosampler for direct vapor equilibration isotope measurements". Your feedback is valuable to us and we will make the recommended minor revision accordingly.

The revision has addressed all comments. There is a minor formatting error in the abstract on line 14 and in the MS line 183 (several instances) - it's unclear regarding "bias" what the number means - for example, a bias of -3.9e-15 ‰ (??), which is nonsense. I suspect a formatting defect.

We changed this to a "-3.9 $\times 10^{-15}$ ‰" format.